# *Clostridioides difficile* specific DNA adenine methyltransferase CamA squeezes and flips adenine out of DNA helix

Jujun Zhou [1,3], John R. Horton [1,3], Robert M. Blumenthal[2], Xing Zhang[1 ✉] & Xiaodong Cheng [1 ✉]

*Clostridioides difficile* infections are an urgent medical problem. The newly discovered *C. difficile* <u>a</u>denine <u>m</u>ethyltransferase <u>A</u> (CamA) is specified by all *C. difficile* genomes sequenced to date (>300), but is rare among other bacteria. CamA is an orphan methyltransferase, unassociated with a restriction endonuclease. CamA-mediated methylation at CAAAA<u>A</u> is required for normal sporulation, biofilm formation, and intestinal colonization by *C. difficile*. We characterized CamA kinetic parameters, and determined its structure bound to DNA containing the recognition sequence. CamA contains an N-terminal domain for catalyzing methyl transfer, and a C-terminal DNA recognition domain. Major and minor groove DNA contacts in the recognition site involve base-specific hydrogen bonds, van der Waals contacts and the Watson-Crick pairing of a rearranged A:T base pair. These provide sufficient sequence discrimination to ensure high specificity. Finally, the surprisingly weak binding of the methyl donor *S*-adenosyl-L-methionine (SAM) might provide avenues for inhibiting CamA activity using SAM analogs.

[1] Department of Epigenetics and Molecular Carcinogenesis, University of Texas MD Anderson Cancer Center, Houston, TX, USA. [2] Department of Medical Microbiology and Immunology, and Program in Bioinformatics, The University of Toledo College of Medicine and Life Sciences, Toledo, OH, USA. [3]These authors contributed equally: Jujun Zhou, John R. Horton. ✉email: XZhang21@mdanderson.org; XCheng5@mdanderson.org

There is an acute medical need to control gastrointestinal infections caused by *Clostridioides difficile* (also known as *Clostridium difficile*)[1–3]. In the developed world, *C. difficile* is one of the leading causes of hospital-acquired infections, often following antibiotic therapy[4]. This pathogen produces endospores, making decontamination difficult[5,6], and it causes deadly infections by producing significant toxins[7,8], and by stabilizing the colon bacterial population (microbiota, or microbiome) in an unhealthy distribution[9,10]. The US Centers for Disease Control and Prevention has classified *C. difficile* infections as being an urgent healthcare risk, associated with substantial morbidity and mortality[11]. Thus far, the medical need is unmet by therapeutic strategies in treatment of *C. difficile* infection and its recurrence[12–15]. Novel targeted therapeutics are urgently needed to combat *C. difficile* infection.

Post-synthetic methylations of DNA are common, and play significant roles, in a wide range of bacterial and archaeal cellular functions[16–19], such as adenine methylation-directed mismatch repair in *Escherichia coli* by Dam[20], and essential functions of the cell-cycle regulated adenine methyltransferase (MTase) CcrM in *Caulobacter crescentus*[21]. The majority of known bacterial and archaeal DNA MTases function to protect a cell's own DNA from digestion by a paired (cognate) restriction endonuclease[22]. However, there are bacterial "orphan" MTases (such as Dam and CcrM)—so named as they are not paired with a restriction endonuclease—that in many cases are involved in controlling chromosome replication, DNA repair, and gene expression[16,23].

The newly discovered CamA enzyme (named for *Clostridioides difficile adenine methyltransferase A*) is another orphan MTase, is present in all *C. difficile* genomes sequenced to date (>300), and is active in all *C. difficile* genomes subjected to PacBio single-molecule real-time DNA sequencing (which can detect N6-methyladenine—N6mA), but is rarely found in other bacteria[24]. CamA-mediated methylation at CAAAA<u>A</u> (underlining indicates the target A) is required for normal sporulation and biofilm production by *C. difficile*, a key step in the disease transmission, as well as for colonization in animal models[24]. Here, we show the kinetic parameters and structural features of CamA in complex with cognate substrate DNA and, given the critical consequences of *C. difficile* infection to human health, discuss the potential for CamA as a novel therapeutic target.

## Results

**CamA exhibits weak binding of SAM.** We purified recombinant full-length CamA [from gene CD2758 of reference strain 630[25]], and measured its enzymatic activity on a double stranded DNA oligonucleotide (oligo) containing a single recognition sequence (Fig. 1). We first optimized the reaction conditions for pH, ionic strength, reaction time, and enzyme concentration (Supplementary Fig. 1). Under the optimal laboratory conditions (pH 7.5, 100 mM NaCl, $t = 2.5$ min), we next measured the CamA kinetic parameters by varying, respectively, concentrations of the DNA substrate and SAM (Fig. 1a, b). CamA has $k_{cat}$ values of 3.9 min$^{-1}$ and 5.4 min$^{-1}$, respectively, for substrate DNA and SAM. For comparison, the measured $k_{cat}$ values for the two other well-studied DNA orphan adenine MTases, under their respective assay conditions, are 0.14 min$^{-1}$ (*E. coli* Dam)[26] and 5.2 min$^{-1}$ (*C. crescentus* CcrM)[27]. However, the CamA binding affinities (as reflected by $K_m$ values) varied from 0.1 μM for DNA to >17 μM for SAM (Fig. 1c). While the $K_m$ value for DNA is comparable to that of *E. coli* Dam (55 nM)[26] and CcrM (17–23 nM)[27], CamA appears to have substantially lower affinity for the methyl donor SAM than that of *E. coli* Dam (3–6 μM)[26,28,29] and phage T4 Dam (0.5 μM)[30]. For context, while we are not aware that SAM levels have been measured in *C. difficile*, in *E. coli* cells the concentration of SAM ranges from ~30 μM during logarithmic growth up to ~230 μM in stationary phase cells[31].

To validate the weaker binding of SAM, we used isothermal titration calorimetry (ITC) to measure the dissociation constants ($K_D$) of CamA with SAM, the reaction product S-adenosylhomocysteine (SAH), and the SAM analog sinefungin (adenosylornithine) (Fig. 1d and Supplementary Fig. 2). Under constant conditions, CamA displayed the weakest binding to SAM ($K_D = 23$–35 μM), followed by sinefungin (19 μM) and SAH (8 μM).

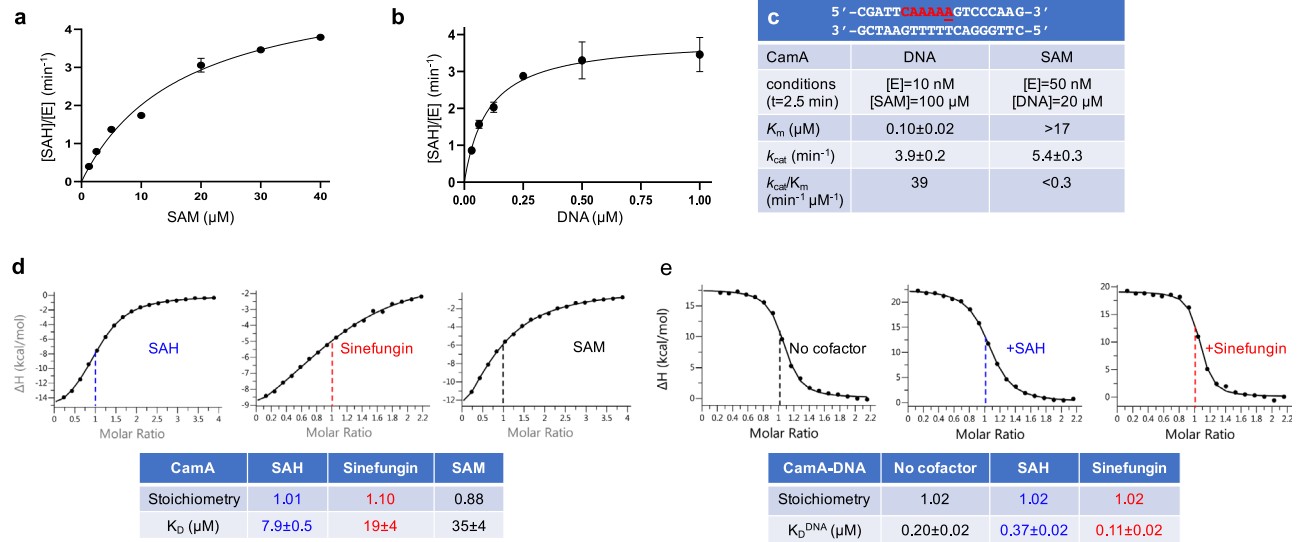

**Fig. 1 Activity of CamA. a**, **b** The formation of byproduct SAH was measured in a bioluminescence assay, by varying concentrations of methyl donor SAM (**a**) or substrate DNA (**b**) ($N = 2$). The dependence of the velocity of product SAH formation per enzyme molecule [SAH]/[E] on substrate concentration was analyzed according to the Michaelis–Menten equation. **c** Summary of CamA kinetic parameters. The DNA substrate used is also shown. Data represent the mean ± SD of two independent determinations, with duplicates assayed for each of the two determinations ($N = 2$). Source data are provided as a Source Data file. **d** ITC measurements of dissociation constants ($K_D$) and stoichiometry of CamA for SAM, SAH, and sinefungin, with $N$ number of independent determinations ($N = 3$ for SAH, $N = 2$ each for SAM and sinefungin) (Supplementary Fig. 2). **e** Influence of cofactor on CamA-DNA binding dissociation (two independent determinations $N = 2$; Supplementary Fig. 3b, c).

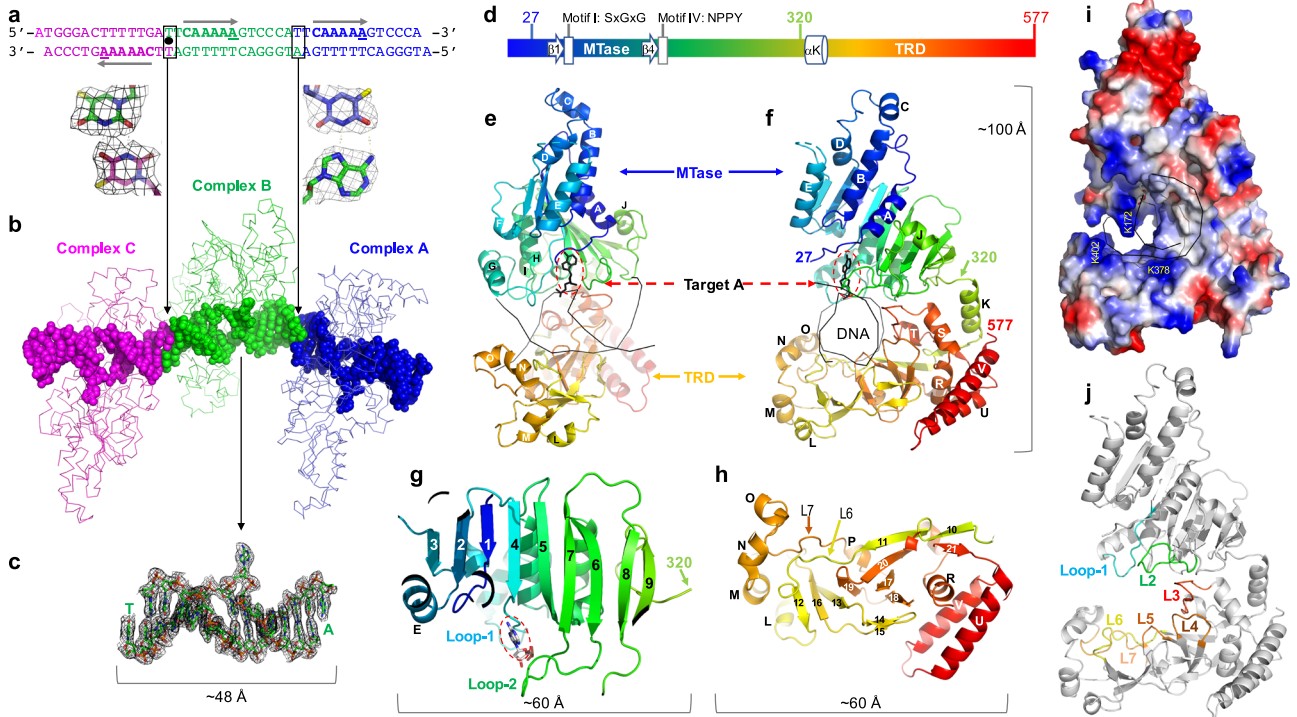

**Fig. 2 Structure of CamA-DNA complex. a** DNA oligos used for co-crystallization are connected by two different joints in the crystals (T:T mismatch linking complexes B and C, and T:A in a Hoogsteen base pair linking complexes A and B). The CamA recognition sites (CAAAA**A**) are indicated by arrows from 5′ to 3′, and the three target adenines are each separated by ~1.5 turns of DNA (14 bp). **b** Three CamA-DNA complexes were formed in the crystallographic asymmetric unit. **c** Omit electron density map (contoured at 4.5 σ above the mean) for the DNA molecule of complex B. The base-flipped adenine is visible. **d** Schematic illustration of CamA with relative locations of motifs I and IV and TRD. Residue 320 before αK indicates the linker point between the two domains. **e**, **f** Two orthogonal views of CamA, colored in spectrum from blue (N-terminus) to red (C-terminus). **g** The N-terminal catalytic domain folds into a nine-stranded sheet. Loop-1 follows strand β4 and loop-2 is between strands β6 and β7. **h** Antiparallel β-structure of C-terminal TRD. **i** Electrostatic surface of CamA with blue for positive and red for negative charges. **j** Seven loops (L1–L7) line the basic surface of the cleft for DNA binding.

The $K_D$ values for SAM and sinefungin are in approximately the same range as the $K_m$ value for SAM (>17 μM). For comparison, M.TaqI has the binding order preference of sinefungin ($K_D$ = 0.34 μM) >> SAM (2.0 μM) > SAH (2.4 μM)[32]. While the actual $K_D$ values were determined under different conditions, the relative binding preferences can be compared, and most strikingly show CamA (unlike M.TaqI) to prefer SAH binding to that of SAM.

Next, we measured the binding affinity of CamA to substrate DNA in the absence and presence of sinefungin or SAH. First, CamA exhibited DNA binding affinity dependent on ionic strength, with $K_D$ value of 40 nM at 150 mM NaCl increasing to ~0.2 μM at 250 mM NaCl (indicating decreased binding; Supplementary Fig. 3a). Second, compared to absence of cofactor, CamA demonstrated about 1.8× stronger binding to substrate DNA in the presence of sinefungin (from 0.20 to 0.11 μM), whereas the presence of SAH yielded about 1.8× weaker binding to DNA (from 0.20 to 0.37 μM; Fig. 1e). The opposite effects of sinefungin and SAH on DNA binding are small but repeatable (Supplementary Fig. 3b, c), and might represent two CamA reaction states (substrate complex before, and product complex after, methyltransfer). Like SAM, sinefungin also carries a formal positive charge, but does not support methyl transfer, and may thus mimic the pre-transfer state.

**Structure of CamA in complex with substrate DNA.** We next sought to understand how CamA recognizes the substrate DNA sequence, and how it specifically methylates the adenine five nucleotides away from the 5′ C:G base pair in CAAAA**A**. Accordingly, we purified CamA-DNA complexes (Supplementary

Fig. 4a, b) and grew co-crystals, using a 13-base pair duplex containing a centrally-located **A**:T and with a 5′-overhang at each end (either an A or T) (Fig. 2a). The complex crystallized in space group $P2_12_12_1$, resulting in a structure determined to a resolution of 2.68 Å (Supplementary Table 1). Although CamA protein contains all nine amino acid motifs conserved in the family of DNA adenine MTases[24,33], molecular replacement was not successful using homolog models. We therefore used experimental phasing, from anomalous signals of selenomethionyl-substituted CamA, for de novo structure determination[34] (Supplementary Table 1 and Supplementary Fig. 4c).

In the crystallographic asymmetric unit, there are three CamA-DNA complexes (A, B and C) (Fig. 2b). The DNA molecules of complexes A and B are stacked head-to-tail with the 5′ overhanging bases forming an A:T base pair in the joint (Fig. 2b). In contrast, the DNA molecules of complexes B and C are rotated 180° in regard to each other, forming a T:T mismatch in the joint of DNA molecules (Fig. 2b). Thus, the three DNA segments form a pseudo-continuous dsDNA molecule, with two target **A** residues located on the same strand, while the third target **A** is located on the opposite strand, in a way representing the presence of methylated CAAAA**A** on both sense and antisense strands in the genomes of *C. difficile* isolates[24]. All DNA base pairs, including the joints were observed, in the electron density (Fig. 2c).

The protein components of the three complexes do not directly contact one another, and are highly similar in conformation (Supplementary Fig. 5). Pairwise comparisons revealed root-mean-square deviations of just 0.3–0.5 Å between ~550 pairs of Cα atoms, and so we describe only complex B hereafter. CamA

comprises an N-terminal catalytic domain (residues 1–320) and C-terminal target recognition domain (TRD; residues 320–577) (Fig. 2d–f). The TRD is the region of DNA MTases responsible for DNA sequence specificity[35,36]. The first 26 residues are disordered in the current structure. The catalytic domain is consistent with structures of Class I MTases[37], though most such MTases include seven β-strands, CamA contains a nine stranded β-sheet (β1–β9 in Fig. 2g, with β8–β9 being the added ones). The structure includes six helices (αA–αE and αJ) located on the front side of the sheet and four helices (αF–αI) on the back side of the sheet, resulting in an open αβα sandwich[37–39] (Fig. 2e, f). As is also typical of Class I structures, there is a central topological switch point between strands β1 and β4, where SAM and the target Ade generally binds at the carboxyl ends of these two parallel neighboring strands. In addition to a characteristic reversed β hairpin (β6 and β7) next to strand β5, CamA contains a second consecutive β hairpin (β8 and β9) at the same end of the sheet (Fig. 2g). In this respect, the structure resembles that of the adenine MTase of the TaqI restriction-modification system (PDB 2ADM)[40–42]. The extra strands make the N-terminal catalytic domain larger, with the longest dimension (~60 Å) comparable to that of the larger TRD domain (Fig. 2g, h), and are stabilized via interdomain interactions.

The N-terminal catalytic domain is connected to the C-terminal TRD domain via a linker centered at residue 320 (Fig. 2f and Supplementary Fig. 6a). The TRD domain is folded into 12 strands (β10–β21), forming a series of antiparallel strands in a β-cluster (Fig. 2h). Four short helices (αL–αO) flank this β-cluster structure on one side, while on the other side of the β-structure

are six more helices (αQ–αV) (Fig. 2h). The basic surface of the cleft between the two domains forms the DNA binding site with residues from both domains approaching DNA from opposite directions (Fig. 2i).

**Distortion of DNA conformation.** The CamA-bound DNA molecule undergoes major distortions at the 6-bp recognition site (labeled as 1–6), while the unrecognized flanking sequence maintains B-form (Fig. 3a). We generated a regular B-DNA model of the same sequence, and superimposed it onto the CamA-bound DNA (Fig. 3a). Several substantial distortions were observed. First, the bound DNA molecule is kinked between base pairs 6 and 7 (the joint between target A6 and the 3′ G7) and bent ~30° (Fig. 3b), resulting in the largest movement of ~23 Å towards one end of the DNA molecule (Fig. 3c). Second, the six recognized base pairs have conformational features of base-pair propeller twist (C1:G1 and A3:T3), base-pair buckle (A4:T4), and base pair-rearrangement (A5:T6) (Fig. 3d). The space left by the rearranged A5 is partially occupied by protein residue Lys172, which forms a hydrogen bond (H-bond) with the orphaned base T5. Third, the π-stacking force in the non-target strand is disrupted by Tyr455, which is wedged between T5–T6 (Fig. 3e). Fourth, the target A6 is flipped completely out of the helix (Fig. 3f).

While the extrahelical positioning of the target A6 base is expected, based on numerous studies with other DNA MTases[42–45], the squeezing between the rearranged A5 and G7—the two nucleotides immediately flanking the target A6—is to date unique to CamA. The three phosphate groups, one 5′ (P_{-1}) and

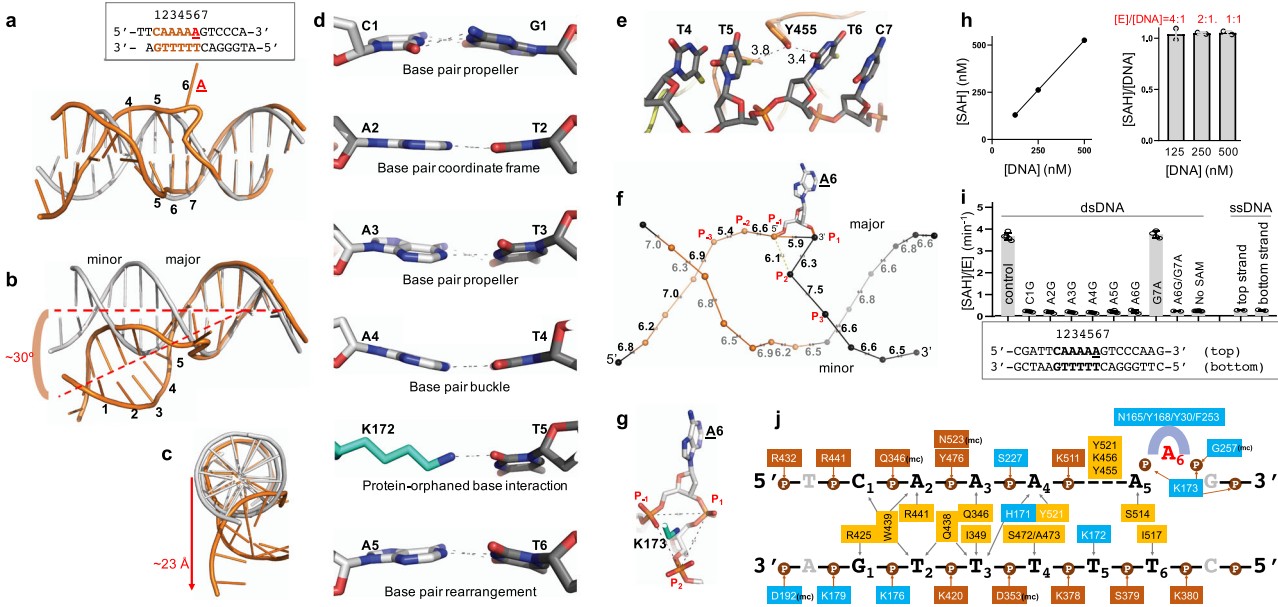

**Fig. 3 CamA-bound DNA conformation. a** The 6-bp recognition sequence is numbered from 1 to 6. B-form DNA (gray) is superimposed with CamA-bound DNA molecule (orange). Note the flipped-out A6 and rearranged base pairs between A5 and T6. **b-c** Conformational deviations of CamA-bound DNA molecule from B-form DNA. **d** Conformational differences among the six base pairs of the recognition sequence. **e** Discontinuous π-stacking between T5 and T6. **f** Inter-phosphate distances of CamA-bound DNA with the largest variations from three phosphate groups 5′ and 3′ respectively to the target adenine (P_{-3} and P_3). **g** Lys173 sits in the center of equilateral triangle of phosphate groups P_{-1}, P_1, and P_2. **h** Under the single turnover conditions where the enzyme is present at or above the DNA substrate concentration, only one methylation event occurs ($N = 2$). Enzyme concentrations used are within a linear range (left). Source data are provided as a Source Data file. **i** Effects of single-base pair substitutions. No activity was observed for ssDNA, using the unsubstituted and unannealed strands separately. Data represent the mean ± SD of $N$ number of independent determinations ($N = 5$ for control, $N = 4$ each for C1G, A2G, A3G, A4G, A6G, and G7A substitutions, $N = 6$ each for A5G and No SAM control, and $N = 2$ each for A6G/G7A and ssDNA substrates) performed in duplicate. Source data are provided as a Source Data file. **j** Schematic illustration of CamA-DNA interactions (for an enlarged version, see Supplementary Fig. 6b): residues in cyan background are from the N-terminal catalytic domain, and residues in (light and dark) orange from the C-terminal TRD. The base-specific contacts are placed between the two strands and the phosphate contacts are depicted above or below the strand. mc, main-chain-atom-mediated contacts.

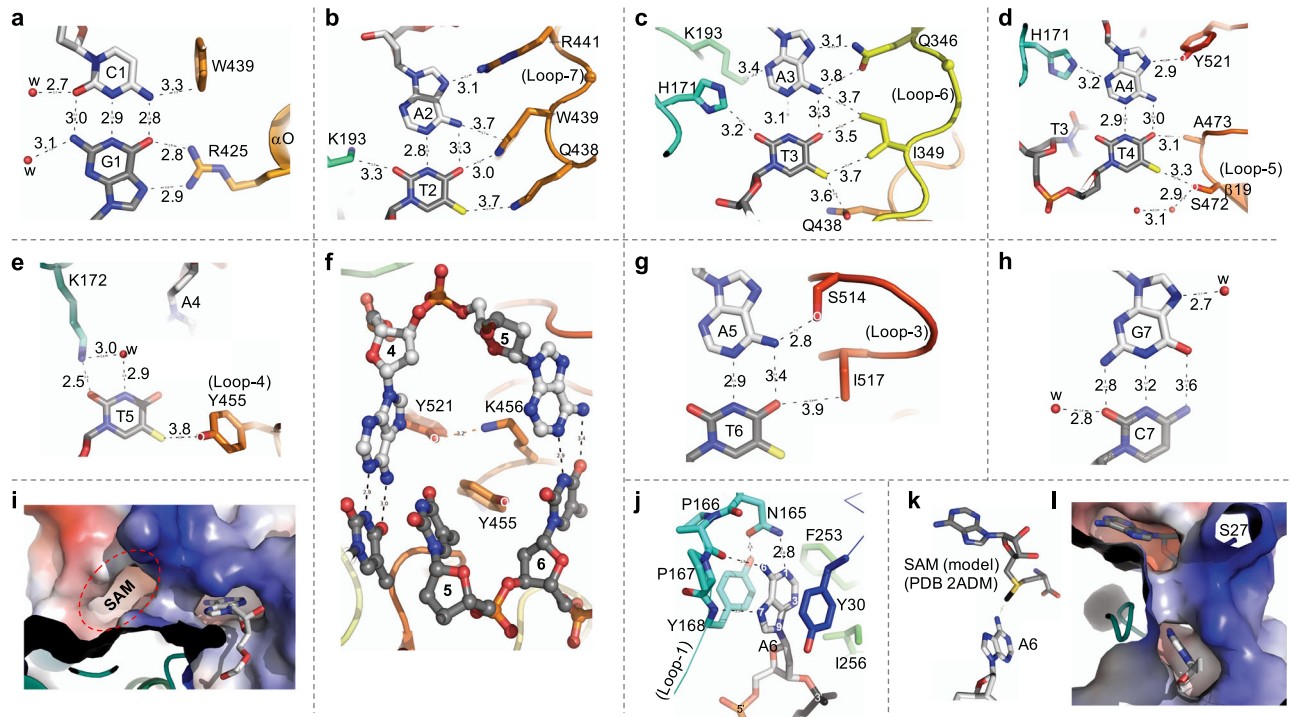

**Fig. 4 CamA-mediated base-specific recognition. a** Interactions with C1:G1 base pair. Interatomic distances are shown in angstroms. **b–d** Interactions with A2:T2, A3:T3, and A4:T4 base pairs at both major (right) and minor (left) grooves. **e** Interactions with the orphan T5. **f** View from the DNA minor groove, with intercalation of TRD residues that occupy the space normally occupied by A5. **g** Interactions with the rearranged A5:T6 base pair. **h** Lack of protein-mediated contacts with G7:C7 base pair. **i** The flipped-out A6 bound in the active-site cage. The nearby SAM-binding pocket is empty in the current structure. **j** Interaction with the target A6 in the aromatic cage. **k, l** A modeled SAM molecule in the cofactor binding pocket. Label S27 indicates the first ordered residue in the current structure.

two 3′ ($P_1$ and $P_2$) to the target A6, yield an equilateral triangular conformation with inter-phosphate distances of ~6 Å (actually 5.9, 6.1, and 6.3 Å; Fig. 3f). These three phosphate groups are balanced by the positively charged Lys173 sitting in the center of the triangle (Fig. 3g), adjacent to the gap-filling Lys172. Currently, we do not know how these events unfold temporally, leading to the observed distortions.

It is interesting that the CamA recognition sequence contains five adenines (CA**A**A**A**A). We first confirmed that, with a DNA substrate having a single occurrence of CAAAAA, only one methylation event occurs under the single turnover conditions with CamA concentration at or higher than that of the DNA substrate (Fig. 3h). We mutated each base pair of the recognition sequence to G:C individually and observed no methyl transfer activity, whereas the substitution of a base pair immediately outside of the recognition sequence had no effect on CamA activity (Fig. 3i).

**Base specific interactions.** Seven loops, two from the catalytic domain and five from the TRD domain, face the DNA, and provide most of the functionally important residues recognizing the six base pairs on both strands (Fig. 2j). There are extensive protein-phosphate interactions, involving many basic residues across both strands of DNA, that form an interface with the 6-bp recognition sequence (Fig. 3j and Supplementary Fig. 6b), and that likely stabilize the distorted protein–DNA complex. This involvement of the catalytic domain in sequence recognition is not that unusual among DNA MTases, as illustrated (for example) by M.SinI via mutagenesis[46] and CcrM by structure[45].

The guanine of the first C1:G1 base pair is recognized by CamA Arg425, which forms bidentate H-bonds with the O6 and

N7 atoms of the G1 base (Fig. 4a), in accordance with the most common mechanism for guanine recognition[47,48]. In addition, the N4 atom of C1 base is in van der Waals contact with Trp439, which in turn contacts the next base pair A2:T2 (Fig. 4b). Adenine A2 forms a H-bond with Arg441 (Fig. 4b), while Gln438 bridges between the two methyl groups of thymine residues T2 and T3 (Fig. 4b, c).

Like G1, adenine A3 is recognized by Gln346 via bidentate H-bonds (Fig. 4c). Juxtaposition of Gln (or Asn) with adenine is a common mechanism for adenine recognition[47], as occurs, for example, with Gln418 of CTCF[49], or Gln264, Asn285 and Gln350 of ZNF410[50]. In such cases, the side chain carboxamide moiety of glutamine or asparagine generally donates one H-bond to adenine N7 and accepts one from adenine N6, respectively, a pattern specific to Ade. However, the side chain of Gln346 is rotated out of the plane of adenine base, with the interatomic distance (3.8 Å) longer between the side-chain oxygen atom and the adenine N6 atom (Fig. 4c). The weakened H-bond interaction is probably due to the neighboring residue Ile349, which sits in the major groove edge of A3:T3 and forms van der Waals contacts with the base pair (Fig. 4c). On the minor groove side, His171 provides a C–H•••O type bond[51] with O2 of T3 base (Fig. 4c). We note that the H-bonding capacity of base pair A3:T3 is fully saturated in the major and minor grooves.

Like A3:T3, the next base pair A4:T4 is also engaged in interactions in both the major and minor grooves (Fig. 4d). Adenine A4 is bordered by His171 (minor groove) and Tyr521 (major groove). Thymine T4 is engaged in two C–H•••O type bonds with Ser472 (interacting with the C5 methyl group) and Ala473 (interacting with the O4 oxygen). Thus, the first four base pairs (C1:G1 to A4:T4) are engaged in extensive direct protein–DNA interactions in both grooves. In addition to single

base-specific interactions, such as Arg425-G1 and Asn346-A3, three residues each bridge between two base pairs, Trp439 between C1 and A2, Gln438 between T2 and T3 and His171 between T3 and A4. In addition, one aromatic (Trp439) and three hydrophobic residues, Ile349, Ala473, and Ile517 (see below), are located intimately at the protein-DNA interface. A tryptophan or histidine spanning two base pairs have been observed previously in ZNF410 and ZFP568, respectively[50,52].

As noted above, the orphaned T5 forms a H-bond via its *O2* atom with Lys172 approaching from the minor groove (Fig. 4e). In addition, Tyr455 approaches from the major groove and makes a weak van der Waals contact with the T5 methyl group. Looking through the DNA, from the minor groove side to the major groove side, two additional protein residues, Lys456 and Tyr521, wedge their side chains into the DNA and push adenine A5 sideways, to the base pairing position initially occupied by target A6, which effectively has been squeezed out (Fig. 4f). The rearranged A5:T6 base pair has two contacts with protein side chains of Ser514 and Ile517 (Fig. 4g); the number of contacts seems to be fewer than what was observed for the first base pairs. As shown by diminished enzymatic activity on substrates having base pair substitution of A5:T5 to G5:C5, a guanine at position 5 would generate a G:T mismatch for the rearranged pair. Beyond the recognition sequence, there is no protein-mediated DNA base contact (Fig. 4h). In sum, the discrimination yielding sequence specificity for CAAAA**A** is provided by base-specific interactions with the first four base pairs of both strands, involving hydrogen bonds and van der Waals contacts in both grooves, and by the base pairing of the rearranged A5:T6. By examining *C. difficile* CamA orthologs that were complete but nonidentical to that from strain 630 (on which this study is based), we note that every identified residue involved in DNA base interactions is completely conserved (Supplementary Fig. 7); so we would expect the sequence specificity to be unchanged as well.

**"Squeezed out" target adenine**. Like other structurally characterized DNA MTases[42–45], the target adenine A6 is flipped out and inserted into an active-site cage (Fig. 4i) formed by three aromatic residues (Fig. 4j). The adenine is stacked in-between Tyr30 and Tyr168, and sealed off with Phe253 in the bottom of the cage (Fig. 4j). Tyr168 is a part of the NPPY motif, a catalytically active-site sequence (motif IV) conserved among amino MTases[33]. The polar groups of the target adenine ring (*N1*, *N6*, and *N7*), that normally form the Watson–Crick pair with thymine and/or interact with protein in the major groove, are now involved in hydrogen bonds with the side chain amide group of Asn165 (interacting with *N1* atom), main-chain carbonyl oxygen of Pro166 (interacting with *N6* amino group), and main-chain amide nitrogen of Tyr168 (interacting with *N7* atom) (Fig. 4j). This pattern of hydrogen bonding defines the specificity for adenine in the active-site binding cage.

**Cofactor-induced conformational change**. Intriguingly, in the CamA-DNA cocrystal structure, we did not observe a bound cofactor in the SAM binding pocket next to the active-site (Fig. 4i), even though exogenous SAH was added during initial complex formation. However, the added SAH had dissociated during subsequent complex purification (Supplementary Fig. 4a, b), probably due to the low binding affinity. Nevertheless, we modeled in a SAM and positioned the methyl group and sulfur atom of SAM in line with the target *N6* atom (Fig. 4k). This linear arrangement, comprising the nucleophile, the methyl group and the leaving thioester group in the transition state, is required for the $S_N2$ reaction mechanism used by SAM-dependent MTases[37]. The model exhibits a good fit, with no apparent clashes (Fig. 4l).

However, nothing in the current model obviously explained the relatively low SAM affinity of CamA.

The absence of cofactor might also explain the disorder of the N-terminal residues (amino acids 1–26), which is near the binding pocket (the first ordered residue S27 is labeled in Fig. 4l). Knowing CamA has weak binding affinity for its cofactor, we repeated the crystallization, this time including SAH throughout the preparation of CamA-DNA-SAH ternary complex. We started with a molar ratio of 1:24 of CamA:SAH, and ended with a final 1:3 ratio of concentrated complex for crystallization (see "Methods" section). The ternary complex was crystallized under conditions similar to those used for the binary CamA-DNA complex, and the structure was determined to a slightly higher resolution of 2.54 Å (Supplementary Table 1). We observed bound SAH in the carboxyl ends of strands β1–β3 (as expected), together with ordered residues of nearly the entire N-terminal fragment starting from residue 2 (Fig. 5a, b). The ordered N-terminal fragment forms two additional helices (residues 9–18 and residues 21–27) and closes off the cofactor binding pocket (Fig. 5c–e). Under some circumstances, this coupled conformational rearrangement might contribute to the low binding affinity of cofactor (SAM, SAH, or sinefungin), though this remains to be tested.

CamA makes extensive contacts with all three moieties of SAH: the adenine ring, the ribose, and the homocysteine. Among the residues interacting with SAH (see below), there are three aspartates, Asp60 of motif I (the last residue of strand β1), Asp114 of motif II (the last residue of strand β2) and Asp149 of motif III (the first residue of helix αF connected to the carboxyl end of strand β3) (Supplementary Fig. 6a). The three negatively charged residues are responsible for binding three moieties of SAH, respectively, the terminal amino group, the ribose hydroxyl oxygen atoms, and the exocyclic amino group of adenine ring.

The adenosyl moiety of SAH forms H-bonds with Asp149 (via its *N6* atom) and the main chain amide nitrogen of Ser150 (via its *N1* atom) (Fig. 5f). In addition, an ethylene glycol molecule (used as the cryoprotectant during crystal freezing) forms two H-bonds with the N6 and N7 atoms, respectively. Furthermore, the SAH adenine ring is stacked between Ile115 and Phe200 (Fig. 5g).

The ribose moiety of SAH engages in five types of interactions. Its hydroxyl groups (2′-OH and 3′-OH) interact with Asp114 (Fig. 5f). The 5-membered ribose ring stacks with the 5-membered pyrrolidine ring of Pro167 (Fig. 5g). The 3′-OH group has a water-mediated interaction with Tyr31 (Fig. 5h). The carbon C2′ is in van der Waals contact with the mainchain carbonyl oxygen atom of Ser27 (Fig. 5h). Fifth among the ribose interactions, the ring O4′ oxygen forms a C–H•••O type interaction with the side chain carbon Cβ atom of Ser62 (Fig. 5i). Ser62 is part of motif I, in the position normally occupied by Phe or Tyr, as discussed below.

Finally, we describe interactions with the aminocarboxypropyl moiety of SAH. This moiety forms an electrostatic interaction with Asp60, H-bonds with Thr32, and van der Waals contact with Tyr31 (Fig. 5j). The sulfur atom of SAH, where a transferable methyl group would be attached in SAM, is in an appropriate distance (3.8 Å) and orientation from the target *N6* atom of Ade of DNA—the methyl acceptor (Fig. 5k). The distance between the sulfur of SAH and the *N6* of target Ade is approximately the sum of the bond distance of donor-methyl ($S^+$-$CH_3$ = 1.82 Å) and the bond distance of acceptor-methyl ($CH_3$-N = 1.5 Å). The H-bonding interactions with the main-chain carbonyl oxygen atoms of Gly28 and Pro166 could facilitate the deprotonation of the target amino group of Ade during catalysis.

Besides binding SAH, the ordered N-terminal residues provided minimal contacts with DNA, with one noticeable exception. Lys25

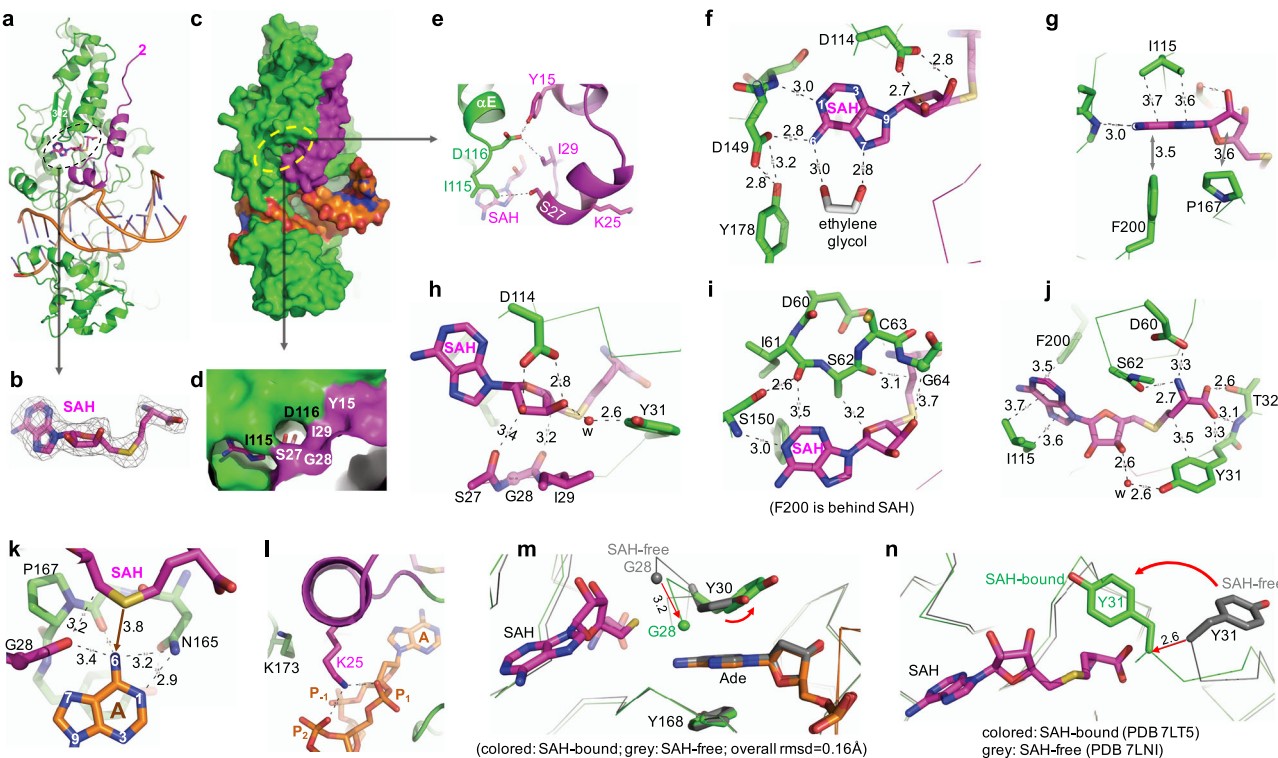

**Fig. 5 Cofactor-induced conformational change. a** The ternary structure of CamA-DNA-SAH (in stick model). The ordered N-terminal residues are in magenta. **b** The omit electron density map (contoured at 5.0σ above the mean) for the bound SAH. **c** Surface presentation of the ternary complex of SAH (in green and magenta), DNA (in orange) and SAH. **d** The enlarged binding pocket of SAH, where the edge of adenine moiety and the ribose hydroxyl groups were visible. **e** Intramolecular interactions between the N-terminal residues (in magenta) and the loop prior to helix αE. **f**, **g** Interactions with SAH adenine moiety. **h** Interactions with SAH ribose moiety. **i** Interactions involving residues of motif I. Note that Phe200 is behind SAH and away from the viewer. **j** Interaction with SAH aminocarboxypropyl moiety. **k** Interactions between the methyl donor (sulfur atom of SAH) and methyl acceptor (N6 of target Ade). **l** Conformational change of Lys25 and Lys173. **m** Superimposition of active sites of complexes B, in the SAH-free (PDB 7LNI) and SAH-bound (PDB 7LT5) states. The bound target adenine is sandwiched between two tyrosine residues, with Tyr30 undergoing a small rotation. **n** The largest movement is Gly28 and Tyr31, which move from an open to the closed conformation upon SAH binding.

interacts with the triangular conformation of three phosphate groups surrounding the target A6 (Fig. 5l). The Lys25–phosphate interaction effectively replaced Lys173, which would be placed in the center of the triangle in the absence of the ordered N-terminal residues (see Fig. 3g). The interplay between the two lysine residues, Lys25 and Lys173, allows CamA to bind DNA even in the absence of bound cofactor, but does not gain additional protein–DNA interactions. While the DNA interface is largely the same in the SAH-free and SAH-bound states, the N-terminal residue Tyr30 involved in binding of the flipped target adenine undergoes a small rotation from the nearly perfect stacking between the two rings of Tyr30 and the target Ade in the SAH-free state to a dislodged angle (Fig. 5m). Accompanying this small movement involved in DNA interaction is the large movement of Gly28 and Tyr31, which close off the SAH binding pocket (Fig. 5n).

**Comparison with other orphan methyltransferases**. Within Class I MTases, CamA belongs to the γ-group[33], based on the sequential order of conserved sequence motifs, particularly sequences for binding the methyl donor SAM (motif I after strand β1) and for catalysis (motif IV after strand β4), and the location of the TRD domain in relation to these two motifs (Fig. 6a, b). As noted, CamA is an orphan MTase, meaning it is not paired with a restriction endonuclease. Examples of such orphan MTases include the DNA adenine MTase (Dam) in *Escherichia coli* (Gammaproteobacteria) and cell cycle-regulated DNA MTase (CcrM) in *Caulobacter crescentus* (Alphaproteobacteria) which

are, respectively, responsible for post-replication maintenance of daughter strand adenine methylation at two very similar sequences: G<u>A</u>TC or G<u>A</u>nTC (n = any nucleotide)[21,53]. Based on the different order of motifs, and the location of the TRD, Dam is a member of α-group and CcrM is a β-group MTase. Taken together, our kinetic and structural characterization of Dam, CcrM, and CamA means that we have characterized an orphan MTase from each group (α, β, and γ)[45,54–56].

Examining the three orphan MTases altogether, there are similarities and major differences among the three enzymes. First, Dam (α group) and CamA (γ group) are active as monomers, having DNA recognition and methylation functions in a single polypeptide, while CcrM (β group) also has both functions in one polypeptide but is only active as a homodimer. The requirement for a dimeric form is unique to the group β MTases due to the relative positions of the functional domains, allowing the enzyme to use division of labor between two subunits in terms of DNA binding and methylation[44,57].

Second, except for the flipped-out target adenine, the Dam-bound DNA conformation has intact intrahelical paired bases. In contrast, CcrM pulls the two DNA strands apart, creating a bubble comprising four enzyme-recognized, unpaired bases, and CamA squeezes out the target adenine by base pair rearrangement (Fig. 6c–e).

Third, all three enzymes use an arginine to interact with a 5′ guanine (Fig. 6f–h) and the Arg-Gua interaction makes sequence-discriminatory contacts in both Dam and CcrM, and most likely also for CamA as shown by the diminished activity of a G1 to

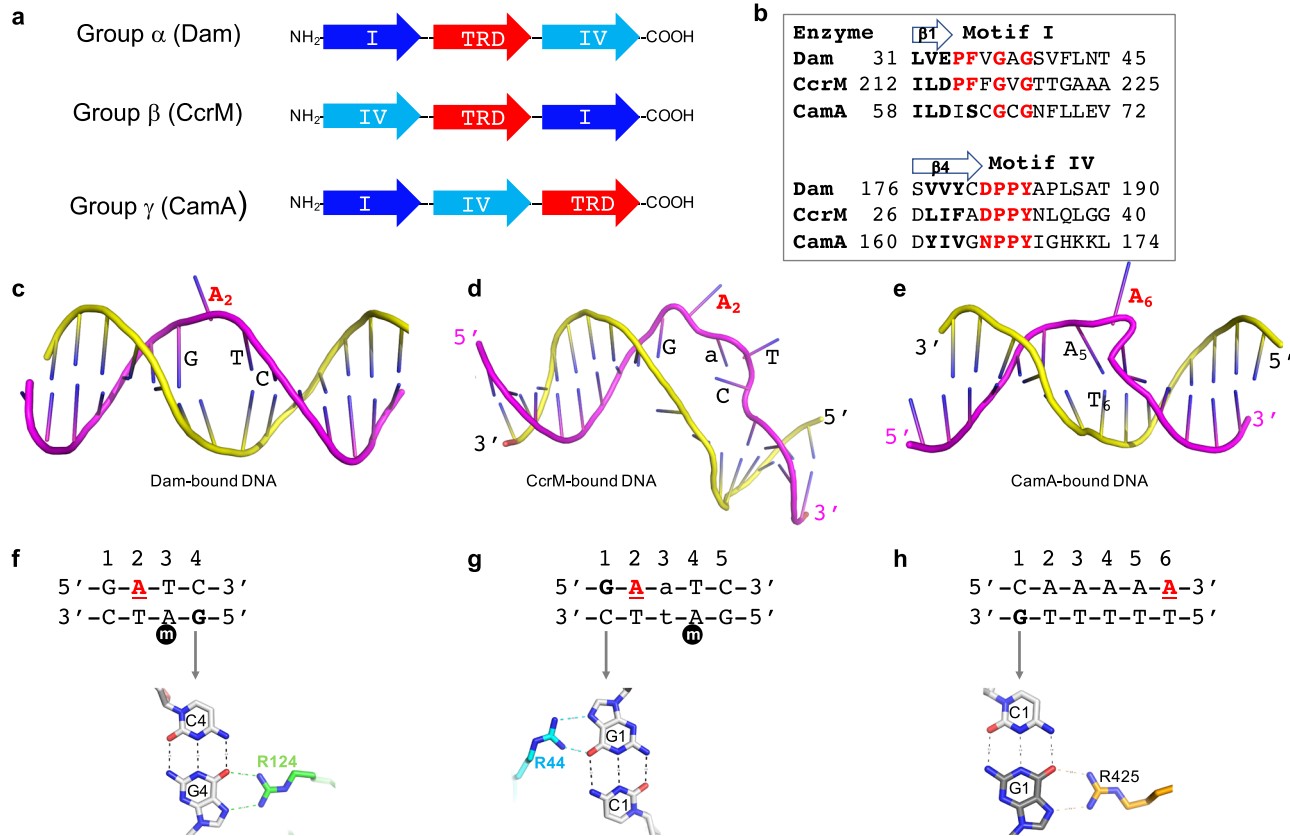

**Fig. 6 Comparison of three orphan adenine methyltransferases. a** Schematic of three Class I groups of amino-MTases, showing altered orders of motifs responsible for SAM binding (motif I), catalysis (motif IV) and target DNA substrate binding (TRD). **b** Sequence alignment of motif I and motif IV for the three orphan adenine MTases. **c–e** Enzyme-bound DNA conformations in Dam (**c**), CcrM (**d**), and CamA (**e**). **f** Dam interacts with guanine G4 of the non-target strand. **g** CcrM interacts with guanine G1 of the target strand. The lowercase m in black circle is the methylated adenine of the parental strand immediately after DNA replication. **h** CamA interacts with guanine G1 of the non-target strand. The underlined A in red in each case is the methylation target.

C1 substitution (see Fig. 3i). However, these contacts are made to different strands relative to the substrate Ade. In Dam and CamA, the recognized Gua is on the opposite strand from the target Ade; whereas in CcrM, the recognized Gua is on the same strand as, and adjacent to, the target Ade.

Fourth, Dam and CamA make base-specific contacts to both DNA strands, whereas CcrM contacts bases only in the target strand[45]. These two features of CcrM, strand separation and base recognition on the same strand that contains the target Ade, allow CcrM (but not Dam) to methylate both double-stranded (ds) and single-stranded (ss) DNA[27]. Like Dam, CamA is inactive on ssDNA (Fig. 3i).

Fifth, the substrates of Dam and CcrM are hemimethylated symmetrical (GATC) or gapped-symmetrical (GAnTC) sequences of newly replicated DNA, and most (if not all) target adenines are located—following replication—on the same (daughter) strand in a uniform direction, allowing the enzymes to be highly processive[26,27,58]. In contrast, CamA methylates an asymmetric 6-bp sequence, and methylation sequence targets in a given chromosome region can occur on either strand[24]. To achieve this, CamA would either slide along the DNA to recognize and methylate the target adenines on the same strand (as illustrated by complexes A and B in Fig. 2a, b); or release from the DNA, turn around, and rebind to methylate the opposite strand (as suggested by comparing complexes B and C in Fig. 2a, b).

Finally, there is one notable difference in the sequence of motif I for binding SAM. In both Dam and CcrM, motif I contains FxGxG (Fig. 6b)—the widespread motif[59] conserved among

many SAM-dependent MTases generating 5-methylcytosine, N4-methylcytosine and N6-methyladenine in either DNA or RNA[33,60,61] as well as in protein glutamine methylation[62]. The phenylalanine (or tyrosine) of motif I provides an edge-to-face interaction to the face of SAM-adenosyl ring, helping to hold it in place. However, in CamA, motif I (SxGxG) has replaced the corresponding phenylalanine with a much smaller Ser62 (Fig. 6b), which interacts with ribose O4′ oxygen (Fig. 5i). This version of motif I is fully conserved among examined CamA orthologs (Supplementary Fig. 7) and, in fact, is a general feature of the γ class of DNA MTases[33]. We now know that the phenylalanine's role can be provided from Phe200 of helix αH (see Fig. 5i for the spatial relationship between Ser62 and Phe200). We note that TaqI MTase too has a motif I (AxAxG) with a small-residue substitution for phenylalanine in the corresponding position. Similarly, the SAM-interacting phenylalanine in M.TaqI is Phe146[32]; the corresponding Phe200 of CamA is very highly conserved among CamA orthologs (Supplementary Fig. 7). Nevertheless, unlike CamA, M.TaqI has an affinity for SAM in the normal range, with a $K_D$ value of 2 μM[32]. We note that another MTase, M.EcoP15I (which has a consensus FxGxG motif), has been crystallized in a complex with DNA, but the cofactor is absent[44].

## Discussion

Here we describe the interaction of *C. difficile* CamA with its cognate DNA substrate, resulting in a rearranged A5:T6 base pair

immediately 5′ to the flipped-out target adenine (A6). The shifted A5:T6 base pair stacks against the 3′ G7:C7 base pair, as if the target A6 was being squeezed out. To be clear, the temporal sequence of events has not yet been determined—whether the "squeezing" precedes and triggers the base flipping, or follows it and prevents the target adenine from flipping back into the double helix is unknown. We note that a shifted base pair has previously been observed for M.HaeIII[63], a DNA cytosine MTase, which methylates the internal cytosine (C3) of GG**C**C. The outer 3′ cytosine (C4) is moved over to form a rearranged base pair with the guanine (G3) that was originally paired with the flipped-out target cytosine. However, there is no apparent base pair stacking with the rearranged C4:G3 base pair in available crystal structures, and thus there is no evidence for "squeezing" as seen in CamA.

We also performed enzymatic analysis of CamA to establish the kinetic parameters, seeking to understand the molecular details underlying CamA catalysis coupled with structural analysis. The observation of weak binding of SAM, together with cofactor-induced conformational change of N-terminal residues near the SAM binding pocket, might provide avenues for inhibiting CamA activity using SAM analogs[64–66]. Iterative cycles of crystallography, synthesis, and bioactivity assays would aid successful design of selective and potent inhibitors of CamA-mediated DNA adenine methylation in *C. difficile*. Such inhibitors could be clinically useful, given that CamA-mediated methylation is required for normal sporulation and colonization by *C. difficile*[24,67]. The profound conformational change when CamA has or lacks a bound cofactor is relatively unique. Further study including mutagenesis is needed to address the mechanism CamA uses for the bi-substrate reaction, and to determine the rate limiting step of the methylation reaction, though some MTases (including phage T4 Dam) are known for which SAH release is the rate-limiting step[68]. It is also possible that the unusually low affinity of CamA for SAM makes it particularly sensitive to in vivo SAM levels, which could play a regulatory role, as has been suggested for MettL16[69,70].

**Is there epigenetic regulation of gene expression in *C difficile*?**. It would be useful to understand in greater depth the molecular basis for the CamA requirement for *C. difficile* sporulation and colonization, though the defect in a Δ*camA* strain appears to be associated with promoters responsive to sigma factors E and F, and the CAAAAA target overlaps with key regulators such as CodY[24,71]. Some other bacterial orphan MTases (not associated with a restriction endonuclease) are involved in chromosome replication, DNA repair, and epigenetic gene regulation[16,23]. For example, the expression of pyelonephritis-associated pili (Pap) in uropathogenic *E. coli* is epigenetically controlled by the methylation state of the two GATC sites (separated by ~90 nucleotides) in the Pap regulatory region[72,73]. In contrast to most GATC sites in the *E. coli* genome, the *pap*-associated sites are not always completely re-methylated after DNA replication, and their methylation state determines in part the phase variation of pilus formation[73]. The failure to methylate these sites is due in part to the binding of regulatory proteins (Lrp, PapI) that block access of Dam[74–76].

If CamA is playing a role in epigenetic gene regulation, one would expect to find examples of unmethylated target sites associated with specific gene regulatory regions. In the 36 SMRT-sequenced *C. difficile* DNA methylomes, there are an average of 7721 occurrences of the CamA recognition sequence, and among them an average of 21.5 (0.3%) CAAAA**A** sites per genome are unmethylated in DNA isolated from log-phase cells[24]. The range varied from 33 genomes containing ~10–30 non-methylated sites

and one genome each at 36, 54, and 152 unmethylated sites (supplementary Table S7A of ref. [24]). Some of these are conserved sites that are unmethylated in a majority of the 36 genomes.

Finally, the role of CamA may help to predict additional features of its kinetic behavior. If the primary role of CamA methylation is regulatory, one would not necessarily expect it to be highly processive, but maybe (like Dam in the control of the Pap locus) responsive to the presence of other regulatory proteins. If, on the other hand, CamA is meant to link the timing of gene expression to replication, as in the case of some *E. coli* transposons responsive to Dam methylation[77–81], it might act more processively on just the daughter strand by following the replication fork, in which case only CAAAAA sites on the leading strand would likely be relevant. In this regard, it is interesting that the presence or absence of sinefungin (a SAM analog) or SAH has little effect (<2×) on CamA's DNA affinity (Fig. 1e). This is consistent with a possible processive mechanism, where SAH–SAM exchange occurs without dissociation from the DNA, though this remains to be tested.

In summary, while it remains to be fully explained why CamA is essential for normal sporulation and gastrointestinal tract colonization by the major pathogen *C. difficile*, it is now clear that this DNA methyltransferase has unusual features that may help in making it a therapeutic target.

## Methods

**CamA gene expression and protein purification**. *C. difficile 630 strain* CamA full length cDNA (gene CD630_27580) was synthesized by Genscript and was cloned into a modified pET-28b vector with a N-terminal His-Sumo tag using a NdeI/BamHI site for creating the expression construct (pXC2184), which was transformed into an *Escherichia coli* Rosetta strain. His-Sumo tagged CamA full length protein was expressed with autoinduction medium[82]. Briefly, an overnight culture grown in 3 mL MDAG medium was inoculated to 3 L ZYM-5052 medium and shaken at 37 °C until the $OD_{600nm}$ reached 0.8. The temperature was then adjusted to 22 °C and cells were cultured overnight to allow autoinduction of target protein. Cells were harvested, resuspended and lysed by sonification in lysis buffer 500 mM NaCl, 20 mM imidazole, 20 mM Tris-HCl pH 8.0, 5% glycerol, 0.5 mM tris(2-carboxyethyl)phosphine (TCEP) supplied with 0.1 mM phenylmethylsulfonyl fluoride (PMSF). The clarified supernatant containing His-sumo tagged CamA was then loaded onto a 5-mL HisTrap™ HP column (GE healthcare), and target CamA protein was eluted with a linear gradient of 20–500 mM imidazole. Fractions containing target protein were pooled together and cut with ULP protease at 4 °C overnight, leaving two additional N-terminal residues (His-Met). The cleaved protein was diluted three times with lysis buffer without NaCl and imidazole and loaded onto a tandem HiTrap Q-SP column (GE healthcare)[83]. After sample was loaded, the Q column was removed, and target protein bound in SP column was eluted with a linear gradient of 0.1–1 M NaCl. Fractions with target protein were pooled and further purified on HiLoad Superdex 200 16/60 (GE healthcare), in buffer 300 mM NaCl, 20 mM Tris-HCl pH 7.5, 0.5 mM TCEP. Fractions with high purity of target protein (judged by SDS PAGE; Supplementary Fig. 1a) were pooled, concentrated to 2.5 mg/mL, flash-frozen in liquid nitrogen and stored in −80 °C.

For expression of L-selenomethionine-substituted CamA, 2.5 mL of an overnight culture in MDAG medium was inoculated into 250 mL adaptable medium containing 0.2× LB, 0.8× M9, 5% glucose, and cultured at 37 °C until $OD_{600nm}$ reached 1.0. Cells were centrifuged, resuspended with 25 mL expression medium containing 1× M9, 0.65% Yeast Nitrogen Base (BD 233520), 5% glucose. A 5-mL of the resuspended culture was inoculated to 0.5 L expression medium and grown in 37 °C until the $OD_{600nm}$ reached 0.8. Temperature was shifted to 18 °C and individual amino acids were added (L-SeMet at 30 mg, Lys, Thr, and Phe at 50 mg, Leu, Ile, and Val at 25 mg). Fifteen min later, target protein was induced with 0.2 mM isopropyl β-D-1-thiogalactopyranoside (IPTG) at 18 °C for 15 h. Purification of SeMet-substituted CamA was the same as the wild type.

**SAM dependent methylation assay**. The DNA methylation activity of CamA was measured by Promega luminescence assay (MTase-Glo™)[84], in which the produced SAH product was converted into ATP in a two-step reaction and the ATP was detected by a luciferase reaction. MTase-Glo luminescence assay produces lower positive false signal compared to other methods[85] and has been employed in our recent studies on SAM-dependent methylation assays[70,86–89]. Typically, for a 10 µL reaction, 2× (CamA and SAM) was preincubated at room temperature (~22 °C) and the reaction was started by adding the same volume of 2× DNA substrate (5′-CGA TTC AAA AAG TCC CAA G-3′ and 3′-GCT AAG TTT TTC AGG GTT C-5′). Reactions were terminated by adding trifluoroacetic acid (TFA) to 0.1 or 0.2% final concentration. The 5 µL of the reaction mixture was added to

low-volume 384-well plate and the luminescence signal was measured by a Synergy 4 multimode microplate reader (BioTek). In reaction buffer containing 20 mM Tris-HCl, 0.1% TFA was used or 0.2% TFA for reaction buffer with 100 mM Tris-HCl to stop the reaction.

For variation of pH (Supplementary Fig. 1c), we used two buffer systems in 50 mM NaCl, 0.1 mg/mL BSA and 1 mM DTT: (1) 10 mM citric acid and 10 mM bis–tris propane (CBTP) for pH 5.5 to 8.8, (2) 20 mM Tris-HCl for pH 7.0 to 8.4, with CamA (50 nM or 100 nM), 20 µM DNA, and 30 µM SAM. Reactions lasted for 10 min.

For variation of ionic strength (Supplementary Fig. 1d), NaCl concentration was varied from 0 to 175 mM NaCl in 20 mM Tris-HCl pH 7.5, 0.1 mg/mL BSA, 1 mM DTT, 50 nM CamA, 20 µM DNA, and 30 µM SAM. Reactions lasted for 10 min.

For variation of the reaction time (Supplementary Fig. 1e), the reaction was carried out with varying time in the optimal buffer (100 mM NaCl, 20 mM Tris-HCl pH 7.5, 0.1 mg/mL BSA, 1 mM DTT) with 25 nM CamA, 20 µM DNA, and 50 µM SAM.

For variation of the enzyme concentration (Supplementary Fig. 1f), reactions were conducted in the optimal buffer with 20 µM DNA and 30 µM SAM for 2.5 min.

To measure the $K_m$ for SAM (Fig. 1a), the reactions were carried out with 50 nM CamA, 20 µM DNA with varying SAM concentration for 2.5 min in the optimal buffer. To measure the $K_m$ for DNA (Fig. 1b), the reactions were 10 nM CamA, 100 µM SAM (~5× above the $K_m$ value for SAM) with varying DNA concentration in the optimized buffer containing 100 mM Tric-HCl for 2.5 min. We note that 100 µM SAM changes the pH of the mixture and 100 mM Tris-HCl is necessary to maintain the pH value around 7.5.

Under the single turnover conditions (Fig. 3h), reactions were carried out overnight with 500 nM CamA, 30 µM SAM, and varying DNA concentration at 125, 250, and 500 nM in the optimized buffer.

For DNA oligos with base pair substitutions (Fig. 3i), reactions were conducted with 50 nM CamA, 5 µM DNA, 40 µM SAM in the optimized buffer for 2.5 min.

**Isothermal titration calorimetry.** All ITC experiments were performed with a MicroCal PEAQ-ITC automated system (Malvern) at 25 °C with reference power of 8 µcal/s. Nineteen injections were performed with an initial injection of 0.2 µL followed by eighteen injections (each of 2 µL) with continuous stirring at 750 rpm. The duration time for the first injection was set at 0.4 s and fixed at 4 s for the following injections and the spacing time between injections was set at 300 s to allow equilibrium.

To measure the binding of CamA to cofactor, SAM, SAH, or sinefungin (450 or 900 µM) was titrated to CamA (40 or 45 µM) in 20 mM Tris-HCl pH 7.5, 250 mM NaCl, 0.5 mM TCEP supplied with 0.9% or 1.8% DMSO (Supplementary Fig. 2). For measuring the binding of CamA to DNA, 200 µM DNA was titrated to 20 µM CamA in 20 mM HEPES pH 8.0, 0.5 mM TCEP, 5% glycerol, 150 mM NaCl, or 250 mM NaCl (Supplementary Fig. 3a). To analyze the effect of presence of cofactor on CamA binding to DNA, 200 µM DNA was titrated to 18 µM CamA with or without 100 µM SAH or sinefungin (Supplementary Fig. 3b). For all ITC experiments, cofactor or DNA oligos in the syringe was titrated to the corresponding buffer as a control and no heat of dilution was detected. The binding data were fitted as "one site" and binding constants were calculated using the ITC analysis module supplied by the manufacturer.

**Purification of protein-DNA complexes for crystallization.** Two complementary DNA oligonucleotides (5′-TTC AAA AAG TCC CA-3′ and 3′- AGT TTT TCA GGG TA-5′) were annealed with 2 mM concentration in 20 mM Tris-HCl pH 7.5, 50 mM NaCl. SAH in stock solution of 25 mM were dissolved in 0.4% HCl. Purified CamA, dsDNA oligo, and SAH were initially mixed in 2 mL at molar ratio 1:1.1:10 (12.5 µM [E], 13.8 µM [DNA], and 125 µM [SAH]) in buffer of 150 mM NaCl, 20 mM Tris-HCl pH 7.5, 0.5 mM TCEP and incubated on ice for 2 h. The 2-mL complex was concentrated by 4× to ~0.5 mL and loaded onto a Superdex 200 Increase 10/300 GL column (GE healthcare) in buffer of 100 mM NaCl, 20 mM Tris-HCl pH 7.5 and 0.5 mM TCEP. Extra DNA molecules were separated and fractions with CamA-DNA complex (Supplementary Fig. 4a) were pooled and concentrated to about 80 µM (~5.2 mg/mL). We note that although SAH was used in the initial complex mixture, the cofactor did not carry through the column.

To prepare the ternary complex of CamA, DNA and SAH, the three components were mixed at concentration of 12.5 µM [CamA], 14.4 µM [DNA] and 300 µM [SAH] in 1.5 mL of 100 mM NaCl, 20 mM Tris-HCl pH 7.5, 0.5 mM TCEP, and incubated on ice for 2 h. The ternary complex was concentrated about 8× to a final concentration of about 100 µM [CamA], 115 µM [DNA], and 300 µM [SAH] and directly used for crystallization.

Crystallization was carried out by an Art Robbins Gryphon Crystallization Robot. Mixture of 0.2 µL complex with 0.2 µL crystallization solution over 70 µL well solution was set up using the sitting drop technique. Single crystals were obtained from solution containing 0.1 M Bis–Tris pH 6.7, 23% polyethylene glycol (PEG) 3350 and 0.3 M potassium citrate. Crystals of SeMet-CamA and DNA complex (without added cofactor) grew in solution containing 0.1 M Tris-HCl pH 7.0, 24.5% PEG 3350 and 0.28 M potassium citrate (Supplementary Fig. 4d). Similarly, the ternary complex of CamA, DNA and SAH were crystallized under conditions of 0.1 M Tris-HCl pH 7.3, 23% PEG 3350, and 0.27 M potassium citrate.

**X-ray crystallography.** When trays with crystallization drops were opened to obtain crystals, some phase separation occurred in the drop and over a short period of time, crystals would tend to dissolve. Thus, crystals were picked up quickly in a nylon loop which were then momentarily placed into mother liquor supplemented with 20% (v/v) ethylene glycol before plunging into liquid nitrogen for cryoprotection. X-ray diffraction data were collected at the SER-CAT beamline 22ID of the Advanced Photon Source at Argonne National Laboratory. For the crystal with the selenomethionyl CamA, X-ray diffraction data was collected 15 eV above the selenium absorption edge (0.97775 Å) and 1000 images were collected rotating the crystal 1° per image so as to obtain large redundancy to achieve an adequate anomalous signal.

The resultant dataset was examined using the Xtriage module of PHENIX[90] which reported a very good anomalous signal to 5.1 Å resolution. The AutoSol module of PHENIX[91] found 15 selenium atoms (five for each protein; Supplementary Fig. 4c) with a Figure-Of-Merit of 0.47 and gave a density-modified map with an R-factor of 0.27. The initial electron density showed recognizable features of secondary structures of β-sheets and α-helices. Reinserting the selenium positions into AutoSol and utilizing the full resolution of the dataset (2.69 Å) gave a very good map in which protein side chains and DNA could be easily identified. In this instance, Autosol reported a Figure-Of-Merit of 0.24 and gave a density-modified map with an R-factor of 0.24. AutoBuild module of PHENIX was utilized to begin the model building which allowed for reiterative processing, and together with manual building in COOT[92], giving improved maps which allowed building of three protein-DNA complexes in the asymmetric unit. COOT was also utilized for corrections between PHENIX refinement rounds.

Data for native CamA crystals were collected at wavelength of 1.00000 Å. The native structures with and without SAH were solved by the difference Fourier method (Supplementary Table 1). In the structure with SAH, difference electron density of bound SAH and the additional residues at the N-termini in the asymmetric unit were immediately obvious and easily built using COOT before refinement. Structure quality was analyzed during PHENIX refinements and finally validated by the PDB validation server[93]. Molecular graphics were generated by using PyMol (Schrödinger, LLC).

**Reporting summary.** Further information on research design is available in the Nature Research Reporting Summary linked to this article.

## Data availability

The experimental data that support the findings of this study are contained within the article. The X-ray structure (coordinates) and the source data (structure factor file) of CamA with bound DNA have been submitted to the PDB under accession numbers 7LNI (SeMet-CamA+DNA), 7LNJ (CamA + DNA) and 7LT5 (CamA + DNA + SAH). The source data underlying Figs. 1a, 1b, 3h, 3l and Supplementary Fig. 1 are provided as a Source Data file with this paper. Source data are provided with this paper.

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

## Acknowledgements

We thank Dr. Clayton Woodcock for his initial involvement in CamA activity studies. We thank Ms. Yu Cao for technical assistance. The work was supported by U.S. National Institutes of Health grant R35GM134744 and Cancer Prevention and Research Institute of Texas grant RR160029. X.C. is a CPRIT Scholar in Cancer Research.

## Author contributions

J.R.H. did cloning and initial purifications and crystallizations. J.Z. performed protein purification, enzymatic assays, crystallization and assisted J.R.H. in X-ray crystallography. R.M.B. participated in discussion, performed bioinformatic analysis, and assisted in preparing the manuscript. X.Z. and X.C. organized and designed the scope of the study.

## Competing interests

The authors declare no competing interests.
