## [Peer Review File · Nature Communications]

REVIEWER COMMENTS

Reviewer #1 (Remarks to the Author):

In this study, Zhou et al characterized the kinetic parameters and structural features of the DNA adenine methyltransferase CamA from *Clostridioides difficile* (*C. difficile*). First, enzymatic and ITC binding assays indicate that CamA has a relatively weak binding affinity for SAM in comparison with other methyltransferases, as well as with SAH and sinefungin. The presence of sinefungin and SAH appears to affect the CamA-DNA binding differently. In addition, CamA shows a strict sequence specificity toward the CAAAAA-containing substrate. Second, structural studies of the CamA-DNA complexes reveal base-specific protein-DNA interactions and base-pair rearrangement at the target site, explaining the high substrate specificity of CamA. Structural comparison of the SAH-free and SAH-bound CamA-DNA complexes further reveals a cofactor binding-induced folding of the N-terminal fragment of CamA, a region that also makes contact with DNA. In addition, structural comparison of CamA with two other DNA adenine methyltransferases, Dam from group α and CcrM from group β , indicates different substrate recognition modes and/or protein assembly states. The activity of CamA is essential for the colonization of *C. difficile*, a leading cause of bacterial infection. In this regard, structure determination of CamA provides a framework for development of new therapeutics targeting *C. difficile*, as well as insight into the mechanisms of DNA adenine methyltransferases. The main concern of this study lies in the fact that some of the biochemical data and the structural observations are not well connected, as detailed below.

Major:

1. One of the major observations of this work is the weak SAM-binding affinity of CamA, in comparison with other methyltransferases as well as with the 4-fold and 1.5-fold stronger affinity for SAH and sinefungin. However, the molecular basis for the differential binding affinities of CamA toward SAM, SAH and sinefungin was neither tested nor analyzed. The authors generated a SAM-bound structural model based on the apo state, rather than the SAH-bound form, the rationale behind which is unclear. Does the cofactor-induced ordering of the N-terminal fragment or the sequence composition of motif I provide an explanation for the relatively weak SAM binding? Note that the comparison of CamA-SAM binding with the reported M.TaqI-SAM binding is unjustified, as the two binding assays involve different experimental conditions. Mutational studies of the cofactor-binding site, combined with enzymatic assays, would help illustrate this important question.
2. The ITC binding assays indicate that the presence of sinefungin leads to a 2-fold increase in CamA-DNA binding, whereas the presence of SAH reduces the CamA-DNA binding by 2-fold. First, the ITC data in Figure 1F need to be repeated to ensure reliable comparison. Note that the titration curve obtained in the absence of cofactor appears to give a different stoichiometric ratio than those determined in the presence of cofactors. Second, these data were not well explained in the context of structural observations. Given that the presence of SAH induces the folding of the N-terminal fragment, which is also a DNA-binding element, it is counterintuitive that the presence of SAH causes the reduction, rather than increase, of the CamA-DNA binding affinity. How is the DNA-binding surface of SAH-free CamA compared with its SAH-bound form?

Minor:

1. The manuscript contains quite a lot of grammatical issues that need to be fixed. To list a few, the sentence "For comparison, the measured k_{cat} values for the two other well-studied DNA orphan adenine MTases, under their respective assay conditions, are *E. coli* Dam (0.14

min⁻¹) and *C. crescentus* CcrM (5.2 min⁻¹)” should be “For comparison, the measured kcat values for the two other well-studied DNA orphan adenine MTases, under their respective assay conditions, are 0.14 min⁻¹ (*E. coli* Dam) and 5.2 min⁻¹ (*C. crescentus* CcrM)” (page 5), “The absence of cofactor might also explain the disordered N-terminal residues (amino acids 1-26) (page 12) should be “The absence of cofactor might also explain the disorder of the N-terminal residues (amino acids 1-26)”, “water-mediated interact with” (page 13), “there is an average of 7,721 occurrences” (page 19), etc.

2. The color scheme for protein residues was used in a way that often obscured the specific atom types. For instance, it is difficult to discern the oxygen atoms involved in the hydrogen bonding interactions depicted in Figure 4G.

3. The discussion “The low binding affinity (i.e., higher value $K_D = K_{off} / K_{on}$) might reflect the lower rate constant for SAH cofactor product dissociating from the enzyme or enzyme-DNA complex when the binding pocket is closed off.” needs to be clarified. Shouldn't the lower dissociation rate constant of SAH cause the opposite effect?

4. To support the structural analysis of the DNA conformation, structural alignment the three CamA-bound DNA molecules in one ASU need to be presented.

5. The statement “We suggest that this is probably due to the low binding affinity of CamA for SAH ($K_D \sim 8 \mu\text{M}$, vs. $\sim 80 \mu\text{M}$ of enzyme concentration used during crystallization; Figure 1D).” needs to be clarified. It is unclear what a protein concentration of $80 \mu\text{M}$ has to do with the missing SAH molecule.

6. F200 in Figure 5I needs to be labeled.

Reviewer #2 (Remarks to the Author):

This study by Zhou J et al describes the structural basis of DNA recognition and N6mA (N6-deoxyadenosine methylation) by *Clostridioides difficile* adenine methyltransferase A (CamA). *Clostridioides difficile* causes gastrointestinal infections and is also responsible for hospital-acquired infections. Requirement of N6mA activity of CamA for sporulation and colonization may qualify this enzyme as a potential drug target.

Here, authors have resolved two crystal structures of CamA bound to its cognate ds DNA with and without SAH, a byproduct of methylation reaction to 2.54 and 2.68Å, respectively. The structure was solved by experimental phasing using the Se-Met labeled protein. They were able to model and refine well the full DNA sequence used in crystallization and the entire protein sequence except for the N-terminus tail (aa 1-26) of CamA enzyme. Interestingly, this region became ordered upon binding of SAH. The conclusions are supported by detailed enzyme kinetics of DNA methylation and extensive structural and biochemical characterization of CamA. Overall, these results and analyses are of high quality. The manuscript is written well, and is very easy to follow.

Although the base-flipping mechanism for N6mA is common, the CamA MTase structure does uncover several very unique features and interesting aspects, including rearranged base pairing between A5 and T6, a different type of “squeezing out” mechanism, two consecutive lysines, K172 and K173 stabilizing the unpaired T5 and equilateral triangle (surrounding the target base) of the phosphate groups, respectively. Moreover, this is perhaps the first example of an enzyme methylating a single adenine (A6) base from a stretch of 5 adenines within a single recognition site (6bp, asymmetrical). The structural basis for such specificity presented here by detailed structural characterization is a very useful addition to the field of nucleic acid (N6mA) MTases. Each of 3 A-T base pairs shows a very different set of interactions with protein from the major groove side. This study also

provides a detailed comparison of CamA (γ group) structure with two other orphan MTases Dam (α group), CcrM (β), and discuss common features and differences in the domain organization/orientation in orphan MTases.

The CamA was characterized very recently (Nature Microbiology 2020). This study is timely and important to the field of DNA methylation by orphan methyltransferase with potential role in epigenetic gene regulation, and infectious diseases.

Minor points:

1. A schematic of the recognition site of CamA is shown in Fig. 3J. This reviewer feels that an additional figure (perhaps supplementary) showing a close-up view of the recognition site (5 base pairs plus unpaired T5 and A6) with their interacting protein residues in stick mode can be helpful.
2. Description of the base (A6) specificity can be expanded a bit (on page 12) as it appears from Fig. 4J that the A6 is sandwiched by Y30 and Y168 through stacking interactions from both sides (aromatic cage as it is referred to in Fig. 4J). Is it a common feature in γ -type MTases? If not, it may suggest an interesting feature in CamA, because the target A in other MTases usually stacks with a single aromatic or hydrophobic amino acid. But here, Y30 from the N-terminus tail seems to provide an additional contact to the target base (in addition to facilitating the SAH binding) to further restrict the movement of the target base to ensure its efficient methylation. Also, does Y30 overlay well in two structures?
3. Figure S5: It would be useful to annotate residue numbers atop the alignment (at an interval of 10) especially those described in the main text as they have important role in orienting the target base (A6) in the catalytic pocket (e.g., K172, Y455), SAH recognition, DNA binding etc.

Reply-to-reviewers:

We thank the two referees and the editor for their time, effort, and constructive comments.

For reviewer 1: we have repeated the ITC experiments as suggested, and the results are the same as before. We observed opposite effects of sinefungin and SAH on DNA binding. The effect is relatively small, but is repeatable. We want to thank you by pushing us to think about the differential binding affinities of the cofactor and its analogs in the context of structural observations.

For reviewer 2: we included two new figure panels (Figure 5M and 5N) in the comparison of unique protein side chains of CamA involved in binding DNA as well as of cofactor.

Below are the point-to-point responses in red.

Reviewer #1 (Remarks to the Author):

In this study, Zhou et al characterized the kinetic parameters and structural features of the DNA adenine methyltransferase CamA from *Clostridioides difficile* (*C. difficile*). First, enzymatic and ITC binding assays indicate that CamA has a relatively weak binding affinity for SAM in comparison with other methyltransferases, as well as with SAH and sinefungin. The presence of sinefungin and SAH appears to affect the CamA-DNA binding differently. In addition, CamA shows a strict sequence specificity toward the CAAAAA-containing substrate. Second, structural studies of the CamA-DNA complexes reveal base-specific protein-DNA interactions and base-pair rearrangement at the target site, explaining the high substrate specificity of CamA. Structural comparison of the SAH-free and SAH-bound CamA-DNA complexes further reveals a cofactor binding-induced folding of the N-terminal fragment of CamA, a region that also makes contact with DNA. In addition, structural comparison of CamA with two other DNA adenine methyltransferases, Dam from group α and CcrM from group β , indicates different substrate recognition modes and/or protein assembly states. The activity of CamA is essential for the colonization of *C. difficile*, a leading cause of bacterial infection. In this regard, structure determination of CamA provides a framework for development of new therapeutics targeting *C. difficile*, as well as insight into the mechanisms of DNA adenine methyltransferases.

Reply: We thank the reviewer for the thorough summary of our work.

The main concern of this study lies in the fact that some of the biochemical data and the structural observations are not well connected, as detailed below.

Reply: We repeated ITC experiments as suggested (below) and discussed the differential binding affinities of cofactor and its analogs in the context of structural observations.

Major:

1. One of the major observations of this work is the weak SAM-binding affinity of CamA, in comparison with other methyltransferases as well as with the 4-fold and 1.5-fold stronger affinity for SAH and sinefungin. However, the molecular basis for the differential binding affinities of

CamA toward SAM, SAH and sinefungin was neither tested nor analyzed. The authors generated a SAM-bound structural model based on the apo state, rather than the SAH-bound form, the rationale behind which is unclear.

Reply: We modeled SAM into the binary structure of CamA-DNA, where the N-terminal residues are disordered, to illustrate that the cofactor (whether it is SAM or SAH) could diffuse in or out of the binding pocket. The N-terminal region appears to provide another layer of regulation, so far unique to CamA, by closing off the cofactor binding site for catalysis, and opening up to allow cofactor exchange. We have added language on p18 to clarify this. On a more speculative note, this unusually high K_M and K_D for SAM may allow CamA activity to be sensitively modulated by intracellular SAM levels, as has been proposed for another MTase, MettL16 (PMID: 28525753 and 33428944. References: 68 and 69).

Does the cofactor-induced ordering of the N-terminal fragment or the sequence composition of motif I provide an explanation for the relatively weak SAM binding?

Reply: We think the cofactor-induced N-terminal conformational change influences the rate of binding (association and dissociation). The motif I residues involved in binding of SAH keep nearly the same conformations in the two structures, with and without bound cofactor. However the other part of the question – whether the sequence of motif I is sufficient to explain the SAM binding – is less clear, and we have not tested it by substitution. As shown in Fig. 6B, and discussed on p.17, the motif I sequences for Dam and CcrM share with that of CamA the presence of a GxG motif, and the highly-conserved D/E 4-residue upstream of the GxG. The most obvious difference is the presence of Phe (the consensus for motif I is FxGxG), where CcrM has a small amino acid (Ser). However, CamA is in group γ , which fairly universally do not have Phe at that position (a spatially equivalent Phe is contributed from αD /motif V in that group – see PMID: 7473738 and Reference 33).

Note that the comparison of CamA-SAM binding with the reported M.TaqI-SAM binding is unjustified, as the two binding assays involve different experimental conditions. Mutational studies of the cofactor-binding site, combined with enzymatic assays, would help illustrate this important question.

Reply: We agree with the reviewer that the two sets of experiments were performed under different conditions, optimized for the respective enzymes. However, we did observe different orders of binding preference of SAH > Sinefungin > SAM for CamA, whereas M.TaqI has the binding order preference of sinefungin >> SAM > SAH. Although we cannot compare the absolute K_D values directly, the relative orders of binding preference should still be informative. As stated in the Discussion, we believe that further study of the differential binding affinities of cofactor should be conjunction with the kinetic mechanism CamA used for the bi-substrate reaction, whether the reaction is processive, and what is the rate limiting step of the methylation reaction. Mutagenesis will be useful to determine the steps along the reaction pathway, but that is outside the scope of the present manuscript (particularly in the absence of clear target residues).

2. The ITC binding assays indicate that the presence of sinefungin leads to a 2-fold increase in CamA-DNA binding, whereas the presence of SAH reduces the CamA-DNA binding by 2-fold.

First, the ITC data in Figure 1F need to be repeated to ensure reliable comparison. Note that the titration curve obtained in the absence of cofactor appears to give a different stoichiometric ratio than those determined in the presence of cofactors.

Reply: We repeated the ITC experiments as suggested, and the results are the same as before. CamA demonstrated increased binding affinity for substrate DNA, by about 1.8X, in the presence of sinefungin (from 0.20 μM to 0.11 μM), whereas the presence of SAH reduced DNA affinity by 1.8X (from 0.20 μM to 0.37 μM) (Figure 1E). Both raw results (old and new) are presented in Figure S3B-C. In addition, we provided all the fitting parameters, including the derived binding affinity (K_D), stoichiometry (the N value=0.88-1.02), entropy and enthalpy of the binding reaction used for generating each fit.

We would emphasize that our ITC experiments have been done multiple times and the results are consistent and repeatable. For example, CamA-SAH binding has been measured three times, and the K_D values are 7.6 μM , 7.9 μM , and 8.1 μM (Figure S2A-C). Similarly, we measured Cam-DNA binding in the absence of cofactor three times, the resultant K_D values are 0.20 μM , 0.18 μM and 0.20 μM , all in 250 mM NaCl (Figure S3).

Second, these data were not well explained in the context of structural observations. Given that the presence of SAH induces the folding of the N-terminal fragment, which is also a DNA-binding element, it is counterintuitive that the presence of SAH causes the reduction, rather than increase, of the CamA-DNA binding affinity. How is the DNA-binding surface of SAH-free CamA compared with its SAH-bound form?

Reply: The ordered N-terminal region does provide Lys25 for contacting DNA. However, this interaction replaces a contact made by Lys173 when the N-terminal region is disordered. Thus, there is no net gain of interactions. As requested by the reviewer 2 (see below), we added a figure panel showing that the N-terminal residue Tyr30, involved in binding of the flipped target adenine, undergoes a small rotation away from the preferred stacking interaction (Figure 5M), but the DNA interface is largely the same with and without bound SAH. Together, this perhaps suggests that CamA is arranged to allow ordering-disordering of the N-terminal region, opening or closing access to the SAM-binding pocket, while making only relatively small changes in the DNA binding affinity (less than 2-fold difference). This may have implications for CamA's ability to methylate processively (allowing SAH-SAM exchange without dissociation from the DNA). Of course, this will need to be tested in the follow-up experiments (discussed on p20).

Minor:

1. The manuscript contains quite a lot of grammatical issues that need to be fixed. To list a few, the sentence "For comparison, the measured k_{cat} values for the two other well-studied DNA orphan adenine MTases, under their respective assay conditions, are *E. coli* Dam (0.14 min^{-1}) and *C. crescentus* CcrM (5.2 min^{-1})" should be "For comparison, the measured k_{cat} values for the two other well-studied DNA orphan adenine MTases, under their respective assay conditions, are 0.14 min^{-1} (*E. coli* Dam) and 5.2 min^{-1} (*C. crescentus* CcrM)" (page 5),

Reply: thank you, we fixed the sentence, and have looked for other problematic sentences.

“The absence of cofactor might also explain the disordered N-terminal residues (amino acids 1-26) (page 12) should be “The absence of cofactor might also explain the disorder of the N-terminal residues (amino acids 1-26)”, “water-mediated interact with” (page 13), “there is an average of 7,721 occurrences” (page 19), etc.

Reply: we thank the reviewer and fixed these grammatical errors. In addition, we went through the entire manuscript and made a number of small changes to wording or punctuation, to further improve readability.

2. The color scheme for protein residues was used in a way that often obscured the specific atom types. For instance, it is difficult to discern the oxygen atoms involved in the hydrogen bonding interactions depicted in Figure 4G.

Reply: We labeled a small “o” on top of the oxygen atom in the panels of Figure 4D (Y521 and S472), 4E (Y455), 4F (Y521 and Y455), and 4G (S514).

3. The discussion “The low binding affinity (i.e., higher value $K_D = K_{off} / K_{on}$) might reflect the lower rate constant for SAH cofactor product dissociating from the enzyme or enzyme-DNA complex when the binding pocket is closed off.” needs to be clarified. Shouldn't the lower dissociation rate constant of SAH cause the opposite effect?

Reply: The reviewer is correct, and we deleted this speculative sentence in the Discussion.

4. To support the structural analysis of the DNA conformation, structural alignment the three CamA-bound DNA molecules in one ASU need to be presented.

Reply: The three CamA-DNA complexes within the crystallographic asymmetric unit were superimposed, with the largest difference in DNA (the end without CamA bound) and in protein (the disordered loop between residues 131-141 away from the protein-DNA interface). These were indicated in New Figure S5.

5. The statement “We suggest that this is probably due to the low binding affinity of CamA for SAH ($K_D \sim 8 \mu\text{M}$, vs. $\sim 80 \mu\text{M}$ of enzyme concentration used during crystallization; Figure 1D).” needs to be clarified. It is unclear what a protein concentration of $80 \mu\text{M}$ has to do with the missing SAH molecule.

Reply: The reviewer is right and we made the sentence simpler: “the added SAH had dissociated during subsequent complex purification (Supplementary Figure S4A-B), probably due to the low binding affinity”.

6. F200 in Figure 5I needs to be labeled.

Reply: In the legend, we stated that Phe200 is behind SAH and away from the viewer. Now we also labeled in the panel (F200 is behind SAH).

Reviewer #2 (Remarks to the Author):

This study by Zhou J et al describes the structural basis of DNA recognition and N6mA (N6-deoxyadenosine methylation) by *Clostridioides difficile* adenine methyltransferase A (CamA). *Clostridioides difficile* causes gastrointestinal infections and is also responsible for hospital-acquired infections. Requirement of N6mA activity of CamA for sporulation and colonization may qualify this enzyme as a potential drug target.

Here, authors have resolved two crystal structures of CamA bound to its cognate ds DNA with and without SAH, a byproduct of methylation reaction to 2.54 and 2.68Å, respectively. The structure was solved by experimental phasing using the Se-Met labeled protein. They were able to model and refine well the full DNA sequence used in crystallization and the entire protein sequence except for the N-terminus tail (aa 1-26) of CamA enzyme. Interestingly, this region became ordered upon binding of SAH. The conclusions are supported by detailed enzyme kinetics of DNA methylation and extensive structural and biochemical characterization of CamA. Overall, these results and analyses are of high quality. The manuscript is written well, and is very easy to follow.

Reply: We thank the referee.

Although the base-flipping mechanism for N6mA is common, the CamA MTase structure does uncover several very unique features and interesting aspects, including rearranged base pairing between A5 and T6, a different type of “squeezing out” mechanism, two consecutive lysines, K172 and K173 stabilizing the unpaired T5 and equilateral triangle (surrounding the target base) of the phosphate groups, respectively. Moreover, this is perhaps the first example of an enzyme methylating a single adenine (A6) base from a stretch of 5 adenines within a single recognition site (6bp, asymmetrical). The structural basis for such specificity presented here by detailed structural characterization is a very useful addition to the field of nucleic acid (N6mA) MTases. Each of 3 A-T base pairs shows a very different set of interactions with protein from the major groove side. This study also provides a detailed comparison of CamA (γ group) structure with two other orphan MTases Dam (α group), CcrM (β), and discuss common features and differences in the domain organization/orientation in orphan MTases.

Reply: We thank the referee.

The CamA was characterized very recently (Nature Microbiology 2020). This study is timely and important to the field of DNA methylation by orphan methyltransferase with potential role in epigenetic gene regulation, and infectious diseases.

Reply: We thank the referee.

Minor points:

1. A schematic of the recognition site of CamA is shown in Fig. 3J. This reviewer feels that an additional figure (perhaps supplementary) showing a close-up view of the recognition site (5 base pairs plus unpaired T5 and A6) with their interacting protein residues in stick mode can be

helpful.

Reply: An enlarged figure is included in the supplement (S6B).

2. Description of the base (A6) specificity can be expanded a bit (on page 12) as it appears from Fig. 4J that the A6 is sandwiched by Y30 and Y168 through stacking interactions from both sides (aromatic cage as it is referred to in Fig. 4J). Is it a common feature in γ -type MTases? If not, it may suggest an interesting feature in CamA, because the target A in other MTases usually stacks with a single aromatic or hydrophobic amino acid. But here, Y30 from the N-terminus tail seems to provide an additional contact to the target base (in addition to facilitating the SAH binding) to further restrict the movement of the target base to ensure its efficient methylation. Also, does Y30 overlay well in two structures?

Reply: It is unique that the target Ade is sandwiched between two Tyr residues, but it is not unusual to have aromatic/hydrophobic stacking interactions on both sides of the Ade. For example, in CcrM, the target adenine is stacked in-between a Tyr and a Thr (Horton et al., 2019), and, in M.TaqI, the target adenine is surround by a Tyr and a Val. In the two structures of CamA, in the absence and presence of SAH, the binding of target adenine changes a little, with the side chain of Y30 undergoing a small rotation. In contrast, the largest movement is G28 and Y31, which close off the SAH binding pocket. We included the comparison in Figure 5M and 5N.

3. Figure S5: It would be useful to annotate residue numbers atop the alignment (at an interval of 10) especially those described in the main text as they have important role in orienting the target base (A6) in the catalytic pocket (e.g., K172, Y455), SAH recognition, DNA binding etc.

Reply: As suggested, we included residues numbers as well as labels of important residues for SAH (s**), target adenine (**a**), and DNA base recognition (**b**), and DNA phosphate interactions (**p**) in the New Figure S7.**

REVIEWER COMMENTS

Reviewer #1 (Remarks to the Author):

The authors have addressed all my concerns. I recommend for publication of this paper in Nature Communications.